# Nurses' 12-hour shifts and missed or delayed vital signs observations on hospital wards: retrospective observational study

Chiara Dall'Ora,[1,2] Peter Griffiths,[1,2] Oliver Redfern,[3,4] Alejandra Recio-Saucedo,[2] Paul Meredith,[5] Jane Ball,[2] the Missed Care Study Group

CD'O and PG contributed equally.

For numbered affiliations see end of article.

**Correspondence to**
Dr Chiara Dall'Ora;
C.Dall'ora@soton.ac.uk

## ABSTRACT

**Objectives** 12-hour shifts worked by nurses on acute hospital wards have been associated with increased rates of missed care reported by nurses. This study aimed to measure the association between nurses working shifts of at least 12 hours and an objective measure of missed care: vital signs observations taken on time according to an acuity-based surveillance protocol.

**Design** A retrospective observational study using routinely collected data from March 2012 to March 2015.

**Setting** 32 general inpatient wards at a large acute hospital in England.

**Participants** 658 628 nursing shifts nested in 24 069 ward days.

**Outcome measures** The rate of daily delayed and missed vital signs observations. We focused on situations where vital signs observations were required at least every 4 hours and measured the number of instances where observations were delayed or missed, per 24-hour period. For each ward and each day, shift patterns were characterised in terms of proportion of care hours per patient day deriving from 'long' shifts (≥12 hours) for both registered nurses and healthcare assistants.

**Results** On 99 043 occasions (53%), observations were significantly delayed, and on 81 568 occasions (44%), observations were missed. Observations were more likely to be delayed when a higher proportion of the hours worked by healthcare assistants were part of long shifts (IRR=1.05; 95% CI 1.00 to 1.10). No significant association was found in relation to the proportion of hours registered nurses worked as long shifts.

**Conclusion** On days when a higher proportion of hours worked by healthcare assistants are from long shifts, the risk of delaying vital signs observations is higher, suggesting lower job performance. While longer shifts are thought to require fewer staff resources to maintain nurse-to-patient ratios, any benefits may be lost if staff become less productive.

## BACKGROUND

Shifts of 12 hours or longer are becoming increasingly common in nursing, based on perceived efficiency savings achieved by reducing the overlap between shorter shifts

### Strengths and limitations of this study

► This study was able to measure an aspect of missed care through an objective outcome, delayed/missed vital signs observations, as opposed to nurse self-reports.
► This study drew on a large dataset of nurses' shifts and patients' vital signs observations and covered 3 years (2012–2015).
► However, the study was carried out in a single hospital.
► Due to the nature of the administrative datasets available, we were not able to control for staff characteristics including age.

associated with a three-shift system.[1] The same average patients-to-nurse ratio could potentially be kept with a lower number of nursing hours, although claims of improved efficiency are yet to be verified in empirical studies. Furthermore, there is no evidence to suggest that organising the delivery of nursing services in acute hospitals with ≥12-hour ('long') shifts improves the safety or efficacy of patient care. A recent literature review found no evidence of a beneficial effect of long shifts on patient outcomes,[2] with findings from a number of studies indicating that greater use of long shifts in hospital wards is associated with a number of adverse outcomes, including reducing the quality of care and patient safety[3 4], higher rates of errors[5 6] and increased mortality.[7]

Recent studies have examined the impact of long shifts on nurses' ability to complete all patient care on a given shift. These studies drew on data from 12 European countries and found that nurses working 12-hour shifts were more likely to report higher levels of care being left undone due to lack of time, a phenomenon generally referred to as 'missed care'.[8] The underlying mechanism for this

association has not been studied, although it has been hypothesised that nurses may need to pace during long shifts in order to maintain enough energy until the end of the shift.[9] This would suggest that nurses' productivity when working 12-hour shifts may be lower, potentially counteracting the expected efficiency savings.

Studies examining the association between long shifts and missed care have relied exclusively on nurse-reported measures, which can result in common methods bias.[10] Although the direct study of worker performance under different shift systems is acknowledged to be challenging,[11] recent expansion in hospital information systems has created new opportunities to study the timely completion of nursing work.[12]

There is now substantial evidence that missed care impacts negatively on patients, with consequences ranging from poor experience of care to increased risk of infection, readmissions and adverse outcomes from undetected physiological deterioration.[13] Timely monitoring and recording of patients' vital signs is key to identify patient deterioration,[14] and electronic recording of these observations provides an opportunity to examine an objective indicator of nurses' job performance, which has not been previously explored.

Using routinely collected data, the aim of this study was to investigate the association between the length of shifts worked by nursing staff and adherence to a vital signs observations protocol.

## METHODS

This was a retrospective observational study using routinely collected data on nursing staff, shift patterns and patient vital signs observations. Data were extracted for all general adult wards (n=32) in an acute care hospital Trust in the South of England over a 3-year period (1 April 2012–31 March 2015). These data were obtained as part of a parent study (ISRCTN registration: 17930973; http://www.isrctn.com/ISRCTN17930973).

### Data sources and variables

Records of worked nursing shifts were extracted from an electronic rostering system implemented hospital wide and directly linked to payroll. Patient data were extracted from the Patient Administration Systems and were linked to the roster data at the ward level. The patient sample included all patients who were on the wards in the days included in the analysis. All data were pseudoanonymised at source before being provided to the research team.

The nursing workforce is composed of registered nurses (RNs) and unregistered healthcare assistants (HCAs), also known as healthcare support workers, or 'nursing support staff'. For each ward for each day, we calculated RNs hours per patient day (RN-HPPD) and HCAs hours per patient day (HCA-HPPD). In each case, this was calculated as staff hours/(patient hours/24); a patient day is equivalent to one patient occupying one bed for a full 24 hours. The proportion of care hours per patient day derived from ≥12-hour shifts was calculated for each day in each ward.

Vital signs observations were derived from a database of records made using the VitalPac system, where nurses record clinical data on hand-held devices at the bedside. These systems comprise the only official record of observations that are maintained on the wards in the study and the use of the system is mandated by Trust policy. The frequency of vital signs observations required by a patient at any given time is determined by a protocol based on the National Early Warning Score (NEWS), which defines when the next observation is due. Detailed description of NEWS can be found elsewhere;[15] in broad terms, it requires that acutely ill patients are observed more often and defines a minimum observation frequency for a given level of patient acuity.

For each set of observations, we obtained the expected time to the next observation (based on the protocol) and classified the subsequent observation as delayed (after 1/3 of the defined interval had elapsed) or missed (after 2/3 of the defined interval had elapsed). This classification was arrived at after extensive discussion, examination of the distribution of delays in taking observations and consultation with patient and professional stakeholders, including staff nurses, senior nurses, physicians and clinical data analysts. The classification of missed observations reflects that an observation taken late, but near to the time a subsequent observation is due, is more clearly fulfilling the second scheduled observation than the first.

According to the hospital escalation protocol, patients are considered at high risk when their NEWS score is 6 or above, requiring observations at least once every 4 hours. Since patients with high NEWS scores are those more likely to die and/or to experience critical deterioration,[16] we considered only observations of patients who exhibited NEWS ≥6. This is the group who are deemed to most require the observations undertaking and whose outcomes are most vulnerable to deterioration if not sufficiently monitored.

### Statistical analysis

The first stage of data analysis involved a descriptive analysis of frequencies and summaries of shift patterns and delayed and missed vital signs observations. The Pearson correlation between the proportion of HPPD deriving from ≥12-hour shifts and missed, and delayed observations was calculated for both RNs and HCAs.

Linear mixed-effects Poisson models were then fit. These models sought to examine the effect of higher proportions of RN-HPPD and HCA-HPPD derived from ≥12-hour shifts on the likelihood of delaying or missing a vital signs observation. Analyses were performed at the ward-day level, and delayed/missed vital signs observations were treated as count outcome; the number of observations due was used as an offset. RN-HPPD and HCA-HPPD were added as control variables. Intraclass correlation coefficients (ICCs) were computed from unconditional random intercept models to describe the between-ward variation of delayed and

missed vital signs observations. The ward level ICC for both delayed and missed observations was 0.37. Due to the high ICC and the data structure (ie, observations aggregated at the ward-day level), ward was included as a random effect in the multivariate models. As missed care is associated with total staffing levels,[17] we included both RN-HPPD and HCA-HPPD in all models as control variables. All analysis was undertaken in the R statistical environment V.3.3.2[18] and mixed effects models were fit using the lme4 package.[19]

### Patient involvement

The parent project was developed with significant input from local public and patient groups who helped shape the original research focus and with a coresearcher who was actively involved in day-to-day project activities including discussion and interpretation of results. As part of the parent study, we undertook a series of consultations with public, patient and clinical experts in order to identify issues that should be considered, including determining an acceptable level of vital signs observations compliance (completing a full set of observations in time). The work was completed in stages so that different groups were approached separately to share views relevant to each individual group. Our approach, codesigned with the parent's study patient and public involvement representative/lay coresearcher, was intended to make the key issues to be discussed tangible for a more lay audience and professional stakeholders such as staff nurses.

### RESULTS

There were 1944 nursing staff members in the sample. 1244 staff members were RNs and 700 staff members were classified as HCAs. Nursing staff worked on average 276 shifts within the 3-year study period. In total, 390 325 shifts were worked by RNs and 268 303 shifts were worked by HCAs. The patient population comprised of 142 046 patients; their mean age was 67; the majority (79%) were emergency admissions, and 50% of the patients had no comorbidities (Charlson comorbidity index=0). The total number of observations in high-acuity patients was 184 628. The total number of delayed observations in high-acuity patients was 99 043 (53%), while the missed observations were 81 568 (44%), nested in 24 069 ward-days. Wards differed as regards size (ie, number of beds), mean HPPD and mean skill mix (ie, the proportion of

RN-HPPD over the total HPPD, calculated as RN-HPPD/ (RN-HPPD+HCA HPPD)). For a detailed description of ward characteristics, please see online appendix 1.

Table 1 summarises the distribution of expected, delayed and missed vital signs observations and of the proportion of HCA-HPPD and RN-HPPD derived from ≥12-hour shifts.

The mean number of delayed observations in each ward-day was 4.1, while on average 3.4 observations were missed on each ward-day. Across the 3 years, on average 46% of HCA-HPPD derived from ≥12-hour shifts, with large variation (IQR 0%–79%) and on average 48% of RN-HPPD derived from ≥12-hour shifts (IQR 0%–82%).

The proportion of RN-HPPD contributed by ≥12-hour shifts was positively correlated with the rate of both delayed (r=0.26) and missed observations (r=0.25). Similarly, the proportion of HCA-HPPD deriving from ≥12-hour shifts was also correlated with delayed (r=0.27) and missed observations (r=0.27). Table 2 reports the association between the proportion of RN-HPPD and HCA-HPPD deriving from ≥12-hour shifts and delayed and missed vital signs observations.

In multilevel models controlling for staffing levels, the proportion of HCA-HPPD derived by ≥12 hour shifts was significantly associated with rate of delayed vital signs observations. An additional 1% of HCA-HPPD derived from long shifts was associated with a 5% increase in the rate of delayed observations compared with days where no long shifts were worked by HCAs (IRR=1.05; 95% CI 1.00 to 1.10). Results for missed observations were similar but not statistically significant. When looking at RNs, the proportion of hours worked through long shifts by RNs was not significantly associated with neither delayed nor missed vital signs observations.

To exclude the possibility that the use of ≥12-hour shifts was associated with increased workload of more acute patients, we ran a linear mixed model to establish how proportions of acutely ill patients (NEWS ≥6) on a ward-day were associated with proportions of RN-HPPD and HCA-HPPD deriving from 12-hours shifts, and we found no association (results available from authors).

### DISCUSSION

This is the first study to explore the relationship between long nursing shifts and an objective measure of missed

| Table 1 | Distribution of missed and delayed observations and proportion of HPPD deriving from long shifts | | | | |
|---|---|---|---|---|---|
| | **Mean** | **Median** | **IQR** | **Min** | **Max** |
| Expected observations per 24 hours in each ward | 7.6 | 5 | 2–10 | 1 | 87 |
| Delayed observations per 24 hours in each ward | 4.1 | 2 | 1–5 | 0 | 56 |
| Missed observations per 24 hours in each ward | 3.4 | 2 | 1–4 | 0 | 51 |
| Percentage of HCA-HPPD derived by ≥12-hour shifts | 46 | 57 | 0–79 | 0 | 100 |
| Percentage of RN-HPPD derived by ≥12-hour shifts | 48 | 61 | 0–82 | 0 | 100 |

HCA-HPPD, healthcare assistant hours per patient day; HPPD, hours per patient day; RN-HPPD, registered nurse hours per patient day.

**Table 2** Multivariable association between the proportions of RN-HPPD, HCA-HPPD deriving from 12 hours or more shifts and delayed/missed vital signs observations

| Shift characteristics | IRR* | 95% CI |
|---|---|---|
| Delayed vital signs observations | | |
| Proportion of RN-HPPD deriving from ≥12-hour shifts | 0.96 | 0.91 to 1.01 |
| Proportion of HCA-HPPD deriving from ≥12-hour shifts | 1.05† | 1.00 to 1.10‡ |
| Missed vital signs observations | | |
| Proportion of RN-HPPD deriving from ≥12-hour shifts | 0.96 | 0.91 to 1.01 |
| Proportion of HCA-HPPD deriving from ≥12-hour shifts | 1.04 | 0.99 to 1.09 |

Random effect: ward.
*Incidence rate ratio (all models include random effect for ward and RN-HPPD/HCA-HPPD).
†Statistically significant at p<0.05.
‡The lower bound 95% CI for proportion of HCA-HPPD deriving from ≥12-hour shifts is 1.0004, and it has been rounded to 1.00 for reporting purposes.
HCA-HPPD, healthcare assistant hours per patient day; RN-HPPD, registered nurse hours per patient day.

care in terms of missed/delayed vital signs observations. We found that delays in obtaining vital signs observations were relatively common for patients assessed as requiring frequent observation. For HCAs, where more of their hours were worked as long shifts, there was a significant increase in delayed vital signs observations. We did not find a significant relationship between delayed/missed vital signs observations and the proportion of RN HPPD worked as part of long shifts.

In the context of a growing body of research suggesting that long nursing shifts on hospital wards are associated with lower quality of care and worse patient outcomes, this finding points to one possible mechanism, as using vital signs observations as an indicator, job performance would appear to be lower for HCAs working longer shifts. This in turn is likely to be a product of the increased fatigue that is frequently reported by those working long shifts.[20 21]

Previous research suggesting that the job performance of nursing staff may reduce when working 12-hours shifts has focused specifically on RNs,[4 9 20] but we found no significant association for this staff group. These studies have relied on general reports of missed care, whereas we focused specifically on vital signs observations. Evidence of the association between staffing levels and missed care shows that, in general, RNs are more likely to report that lack of time causes them to miss interpersonal care (eg, talking and comforting patients and planning patient care) than 'clinical' care.[17] This may reflect a prioritisation of some aspects of care by RNs in the face of limited capacity. It might be that other aspects of RN work are adversely affected by 12 hours shifts, but observation

of patient groups identified as at risk of deterioration remains a priority.

The monitoring of vital signs is increasingly being delegated to HCAs. Research undertaken with RNs and HCAs in the UK found that vital signs observations were a central feature of HCAs' roles.[22] Furthermore, there are reports of RNs considering the assessment of vital signs a basic task and therefore one that RNs may delegate to less qualified staff members.[23] The extent to which such delegation has occurred in the present Trust is unclear, as the record of who undertook the observation was incomplete and unreliable, although all HCAs are expected to have the competency to undertake and record vital signs observations. Being unable to establish whether such delegation occurs prevents us from understanding the role HCAs may play in avoiding nursing missed care. Anecdotal accounts suggest that HCAs undertaking vital signs is widespread. Thus, it may simply be that compliance with the vital signs observation protocol is a good indicator of job performance for HCAs but not for RNs because of different responsibilities. Alternatively, it may be that long shifts adversely affect HCAs but not RNs or that RNs place a higher priority on ensuring timely observation even though their overall capacity is similarly affected by long shifts.

This study has some limitations. We were only able to focus on a single aspect of job performance—vital signs monitoring—that may not be reflective of global performance. We were also unable to ascertain which staff group was assigned to take observations. Furthermore, it was not possible to determine whether observations were deliberately omitted for clinical reasons despite being in contravention of the protocol. For example, in a previous study, nursing staff have reported choosing not to perform some observations at night because they felt waking patients could have a negative effect on individual patients. These reasons included patients who had difficulty sleeping and 'confused' patients who may become agitated.[24] Furthermore, patients may not have been available at the time observations were due, for example, if they were undergoing diagnostic tests (eg, radiology) elsewhere in the hospital. We did not study the direct consequences of missed vital signs observations, although other studies have shown that missed care in general is associated with adverse patient outcomes.[13] Recent studies have shown that nurse-reported missed care mediates the relationship between low staffing and mortality,[25] with a further study showing specifically that missed vital signs observations mediated an association between RN staffing levels and mortality.[26]

We were not able to account for staff characteristics including age and personal commitments external to work, both of which may influence job performance. Furthermore, in common with several previous missed care studies,[17] we were not able to obtain a nurse-reported reason for missing vital signs observations. This study

was conducted in a single site, and these findings may not generalise to other National Health Service Trusts. However, a previous study found that shift patterns tend to vary more within hospitals than between hospitals in England.[4]

## CONCLUSIONS

This is the first study in nursing to use objective measures of shift patterns and missed care obtained from records spanning a long period of time. Where working hours involved a larger proportion of long shifts, delays in obtaining scheduled vital signs observations by nursing support staff were observed. While this study lends partial support to previous findings using global self-report measures, further research should explore better measures of job performance and more complete data on costs, and effects, to work out which shift patterns are most cost-effective for ward-based services.

**Author affiliations**
[1]National Institute for Health Research Collaboration for Leadership in Applied Health Research and Care (Wessex), University of Southampton, Southampton, UK
[2]School of Health Sciences, University of Southampton, UK
[3]School of Computing, University of Portsmouth, Portsmouth, UK
[4]Nuffield Department of Clinical Neurosciences, Kadoorie Centre for Critical Care Research and Education, University of Oxford, Oxford, UK
[5]Research and Innovation Department, Portsmouth Hospitals NHS Trust, Portsmouth, UK

**Acknowledgements** The authors thank Portsmouth Hospitals NHS Trust for allowing us to use the study data.

**Collaborators** The Missed Care Study Group comprises Karen Bloor, University of York; Dankmar Böhning, University of Southampton; Jim Briggs, University of Portsmouth, Centre for Healthcare Modelling and Informatics; Anya De longh, Independent lay researcher; Jeremy Jones, University of Southampton, Faculty of Health Sciences; Caroline Kovacs, University of Portsmouth, Centre for Healthcare Modelling and Informatics; David Prytherch, University of Portsmouth, Centre for Healthcare Modelling and Informatics and Portsmouth Hospitals NHS Trust, Clinical Outcomes Research Group; Paul Schmidt, National Institute for Health Research Collaboration for Applied Health Research and Care (Wessex) and Portsmouth Hospitals NHS Trust, Clinical Outcomes Research Group; Nicky Sinden, Portsmouth Hospitals NHS Trust, Clinical Outcomes Research Group; and Gary Smith, Bournemouth University.

**Contributors** CD drafted the article, giving substantial contributions to its conception and design, collaborated in acquiring the data, performed statistical analyses and interpreted the data, and gave final approval of the version to be published. She agreed to be accountable for all aspects of the work in ensuring that questions related to the accuracy or integrity of any part of the work are appropriately investigated and resolved. PG gave substantial contributions to the conception and design of the work, the interpretation of data and revised critically the article for important intellectual content and gave final approval of the version to be published. OR gave substantial contributions to acquiring the data, to the statistical analysis and data interpretation, and revised critically the article for important intellectual content and gave final approval of the version to be published. AR-S collaborated in the conception of the study, revised critically the article for important intellectual content and gave final approval of the version to be published. PM gave substantial contributions to acquiring the data, collaborated with statistical analysis and data interpretation, revised critically the article for important intellectual content and gave final approval of the version to be published. JB gave substantial contributions to the conception of the study, revised critically the article for important intellectual content and gave final approval of the version to be published.

**Funding** This work was supported by National Institute for Health Research Collaboration for Leadership in Applied Health Research and Care Wessex and by the NIHR Health Services & Delivery Research – grant number: 13/114/17: 'Nurse staffing levels, missed vital signs observations and mortality in hospital wards: modelling the consequences and costs of variations in nurse staffing and skill mix'.

**Disclaimer** The views expressed in this publication are those of the authors and not necessarily those of the NHS, the National Institute for Health Research or the Department of Health and Social Care.

**Competing interests** None declared.

**Patient consent** Not required.

**Ethics approval** Ethical approval to undertake this research was granted from the University of Southampton Ethics Committee (Submission Number 18311).

**Provenance and peer review** Not commissioned; externally peer reviewed.

**Data sharing statement** Due to the nature of this study and the conditions attached to original data agreements, there are no data available for wider use. All queries should be submitted to the corresponding author in the first instance.

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
