## [Reviewer comments · BMJ Open]

ARTICLE DETAILS

TITLE (PROVISIONAL)	Nurses' 12-hour shifts and missed or delayed vital signs observations on hospital wards: retrospective observational study
AUTHORS	Dall'Ora, Chiara; Griffiths, Peter; Redfern, Oliver; Recio-Saucedo, Alejandra; Meredith, Paul; Ball, Jane

VERSION 1 – REVIEW

REVIEWER	Nancy M. Daraiseh Cincinnati Children's Hospital, United States
REVIEW RETURNED	04-Jul-2018

GENERAL COMMENTS	The authors undertake an increasingly important topic of missed care and its particular relationship with shift length. In my opinion this paper has the potential to provide a significant contribution to the literature, however it has several limitations that hinder my recommendation for publication. • There is no description of the sample, for RNS or HCAs, patient population, or the wards (of which there are 32!). This is also apparent in the analyses. These factors are important to understand why missed care is occurring. Further, there is no explanation as to why this information was not included.• In Table 2 the 95% CI includes '1' yet the authors conclude the result is statistically significant.• The authors state that the responsibility for obtaining vital signs is increasingly placed on HCAs, which may be a factor in missed care by RNS (i.e. delegation of tasks to HCAs). Since these occurrences are not collected, results may be skewed due to the lack of information.• Although the method for data collection may be objective (i.e. not reliant on nurse report) it is also incomplete, since authors cannot determine if another provider completed the task in the RNs place (e.g. "unable to ascertain which staff group was assigned to take observations" pg 11, line 6). Further no determinations as to why the missed care occurred, e.g. interruptions, emergencies, etc. are provided.• How did the authors develop the missed care classification? Was this based on prior studies?• Minor comments:o On page 7, line 9 the term 'four-hourly' is unclear. Is this to mean every four hours or four, hourly observations, or four every hour?o The 'Patient Involvement' section is vague.
--

REVIEWER	Nicole Blay Western Sydney University, Australia
REVIEW RETURNED	07-Aug-2018

GENERAL COMMENTS	An interesting study that adds to the increasing knowledge around working longer hours and missed care. A couple of minor queries/considerations: The method of observation needs clarifying. Do you mean that the observations were undertaken by nurses and observers recorded their activity at the bedside using the VitalPac system or were charts retrospectively examined using the VitalPac system to extract the data? I am interested to know if the 12hr shift formed a component of the regular shift rostering system or if longer shift hours were in response to staffing/patient acuity that may have had an impact on fatigue? What was the standard skill-mix? Was there evidence of RNs being substituted by HCAs working longer hours? In addition to RNs choosing not to disturb patients it should be considered that patients may not have been available at the time observations were due.
---

VERSION 1 – AUTHOR RESPONSE

Reviewer 1

Reviewer’s comment	Response	Action taken
There is no description of the sample, for RNS or HCAs, patient population, or the wards (of which there are 32!). This is also apparent in the analyses. These factors are important to understand why missed care is occurring. Further, there is no explanation as to why this information was not included.	We agree that a better description of the sample is needed -- thanks for highlighting this. We had originally omitted this information because the multilevel analysis occurs at the ward-day level, rather the nurse level. We have now added a more detailed description of shifts, patients and wards. We would like to highlight that demographic information on nursing staff are not available, as the electronic rostering database we used does not include any demographic information. This is something we clarify in the limitations section of the manuscript.	 - Information on RNs and HCAs has been added at the beginning of the results section at page 8: “There were 1944 nursing staff members in the sample. 1244 staff members were registered nurses and 700 staff members were classified as healthcare assistants. Nursing staff worked on average 276 shifts within the three-year study period” - Information on the patient population has been added at the beginning of the results section at page 8: “The patient population comprised of 142,046 patients; their mean age was 67; the majority (79%) were emergency admissions, and 50% of the patients had no comorbidities (Charlson comorbidity index = 0).” - Detailed information on the 32 wards has been

		added as a supplementary table (Appendix 1). At page 8 we wrote: "Wards differed as regards size (i.e. number of beds), mean HPPD and mean skill mix (i.e. the proportion of RN-HPPD over the total HPPD, calculated as $RN-HPPD / (RN-HPPD + HCA-HPPD)$). For a detailed description of ward characteristics, please see Appendix 1."
In Table 2 the 95% CI includes '1' yet the authors conclude the result is statistically significant.	The exact p value was 0.04 (<0.05) hence statistically significant. The lower limit of the IRR is shown as 1.00 due to rounding. The exact is 1.0045. These results are therefore consistent. We prefer not to over represent precision by reporting results to an additional decimal point although we are happy to be guided by the journal editors on this.	No changes
The authors state that the responsibility for obtaining vitals signs is increasingly placed on HCAs, which may be a factor in missed care by RNS (i.e. delegation of tasks to HCAs). Since these occurrences are not collected, results may be skewed due to the lack of information.	We agree that being unable to capture the extent to which vital signs are delegated to HCAs is a limitation, and we have now acknowledged this in our limitations (pages 10-11). However, we believe that this does not lead to skewed results in the analysis – we measured the association between the proportion of HPPD deriving from 12-h shifts worked by HCAs and vital signs observations, and this association exists regardless of whether delegation occurs.	At page 11 we wrote: "Being unable to establish whether such delegation occurs prevents us from understanding the role HCAs may play in avoiding nursing missed care."
Although the method for data collection may be objective (i.e. not reliant on nurse report) it is also incomplete, since authors cannot determine if another provider completed the task in the RNs	We share the reviewer's concern that our indicator is not complete, and we highlighted it on page 11. However, if a vital signs observation was classified as "missed", it means that no one	We added at page 12: "Furthermore, in common with several previous missed care studies, we weren't able to obtain a nurse-reported reason for missing vital signs observations."

place (e.g. “unable to ascertain which staff group was assigned to take observations” pg 11, line 6). Further no determinations as to why the missed care occurred, e.g. interruptions, emergencies, etc. are provided.	performed the task, neither RNs, HCAs nor doctors – it was missed. Unfortunately, we did not have access to data as to why care was missed. This is very similar to previous missed care studies, where nurses are surveyed without asking them “why was care missed”. The only exception being the MISSCARE survey by Kalisch. What we endeavoured to do in this study was to establish whether there was an association between 12-h shifts and missed/delayed vital signs, controlling for staffing levels, which have been indicated in the literature as being associated with missed care. We have now added in the limitations’ section that our study did not capture the nurse-reported reason for care being missed.	
How did the authors develop the missed care classification? Was this based on prior studies?	The classification of missed (not recorded within the scheduled interval $+> 2/3$ of the interval) observations was created ad hoc for the study based on consensus between expert researchers and stakeholders. We have now added this information.	At page 7 we added: “This classification was arrived at after extensive discussion, examination of the distribution of delays in taking observations and consultation with patient and professional stakeholders (see below). The classification of missed reflects that an observation taken late, but near to the time a subsequent observation is due, is more clearly fulfilling the second scheduled observation than the first ”
On page 7, line 9 the term ‘four-hourly’ is unclear. Is this to mean every four hours or four, hourly observations, or four every hour?	We apologise for the lack of clarity of the term “four-hourly”. We have now changed it to “every four hours”	At page 7 we changed the sentence to: “requiring observations at least once every four hours”.
The ‘Patient Involvement’ section is vague.	We added more detail of the PPI role in the study.	At page 8 we added: “As part of the parent study, we undertook a series of consultations with public, patient and clinical experts in order

		to identify issues that should be considered, including determining an acceptable level of vital signs observations compliance (completing a full set of observations in time). The work was completed in stages so that different groups were approached separately to share views relevant to each individual group. Our approach, co-designed with the parents study's PPI representative / lay co-researcher, was intended to make the key issues to be discussed tangible for a more lay audience and professional stakeholders such as staff nurses."
--	--	---

Reviewer 2

Reviewer's comment	Response	Action taken
The method of observation needs clarifying. Do you mean that the observations were undertaken by nurses and observers recorded their activity at the bedside using the VitalPac system or were charts retrospectively examined using the VitalPac system to extract the data?	We apologise if this was unclear, we have revised it and hope the method of observation is now clearer.	At page 6 we wrote: "Vital signs observations were derived from a database of records made using the VitalPac™ system, where nurses record clinical data on hand held devices at the bedside. These systems comprise the only official record of observations that are maintained on the wards in the study and the use of the system is mandated by Trust policy."
I am interested to know if the 12hr shift formed a component of the regular shift rostering system or if longer shift hours were in response to staffing/patient acuity that may have had an impact on fatigue?	Thank you for highlighting this important mechanism. We can say with certainty that 12-h shifts form a component of the regular shift rostering. We have explored your question re: staffing levels in another publication which is currently in press – we essentially found that increases in proportions of 12-h shifts are not associated with a decrease in staffing levels, even when taking patient acuity into account. In this paper, we controlled for staffing levels (HCA-HPPD and RN-HPPD), meaning that we are	At page 10 we added: "To exclude the possibility that the use of ≥12-hour shifts was associated with increased workload of more acute patients, we ran a linear mixed model to establish how proportions of acutely ill patients (NEWS ≥6) on a ward-day were associated with proportions of RN-HPPD and HCA-HPPD deriving from 12-hour shifts, and we found no association (results available from authors)."

	confident that when the proportion of 12-h shifts changes, variation in staffing levels is also accounted for. To address your concern about patient acuity, we further analysed our data and looked at the association between use of 12-h shifts and patient acuity, and we did not find an association. We added this further analysis to the results section.	
What was the standard skill-mix? Was there evidence of RNs being substituted by HCAs working longer hours?	We added a supplementary table where we reported the ward-by-ward skill mix. There is no evidence of RNs being substituted by HCAs working longer hours, as the percentage of HPPD worked by HCAs derived by 12-h shifts was marginally smaller compared to RNs (46% vs 48%).	At page 8 we wrote: "Wards differed as regards size (i.e. number of beds), mean HPPD and mean skill mix (i.e. the proportion of RN-HPPD over the total HPPD, calculated as $RN-HPPD/(RN-HPPD + HCA-HPPD)$). For a detailed description of ward characteristics, please see Appendix 1
In addition to RNs choosing not to disturb patients it should be considered that patients may not have been available at the time observations were due.	Unfortunately, we did not have information on when patients were away from the ward, but we added this in the limitations' section.	At page 12 we wrote: "Furthermore, patients may not have been available at the time observations were due, for example if they were undergoing diagnostic tests (e.g. radiology) elsewhere in the hospital."

VERSION 2 – REVIEW

REVIEWER	Nancy M Daraiseh Cincinnati Children's Hospital, United States
REVIEW RETURNED	24-Oct-2018

GENERAL COMMENTS	The authors have made the modifications to the last submission and greatly improved the manuscript. Minor modifications:  • In Table 2 the 95% CI includes '1' because authors state the value is 1.0004. Placing this info in a footnote to remove confusion may be helpful while leaving as is cause for speculation. • Page 7, paragraph 2: Can the authors detail who the 'professional stakeholders' are that participated in determining
--

	missed care classifications? Were they, physicians, nurses, HCAs, families? • Page 8, Patient Involvement section: please define the acronym 'PPI'
--	---

REVIEWER	Nicole Blay Western Sydney University, Australia
-----------------	---

REVIEW RETURNED	20-Oct-2018
-------------

GENERAL COMMENTS	The authors have comprehensively addressed the reviewers comments
---

VERSION 2 – AUTHOR RESPONSE

Reviewer 1

Reviewer's comment	Response	Action taken
In Table 2 the 95% CI includes '1' because authors state the value is 1.0004. Placing this info in a footnote to remove confusion may be helpful while leaving as is cause for speculation.	We agree that what you suggest reduces confusion, we have updated table 2.	At page 10, Table 2 we added: "The lower bound 95% confidence interval for proportion of HCA-HPPD deriving from ≥12-h shifts is 1.0004, and it has been rounded to 1.00 for reporting purposes"
Page 7, paragraph 2: Can the authors detail who the 'professional stakeholders' are that participated in determining missed care classifications? Were they, physicians, nurses, HCAs, families?	Thanks, we have added this information.	At page 7 we added: "...including staff nurses, senior nurses, physicians, and clinical data analysts."
Page 8, Patient Involvement section: please define the acronym 'PPI'	Thank you, we added PPI definition.	At page 8 we added: "Patient and Public Involvement (PPI)"